# Risk Factors for Venous Thromboembolism in Severe COVID-19: A Study-Level Meta-Analysis of 21 Studies

**DOI:** 10.3390/ijerph182412944

**Published:** 2021-12-08

**Authors:** Hervé Lobbes, Sabine Mainbourg, Vicky Mai, Marion Douplat, Steeve Provencher, Jean-Christophe Lega

**Affiliations:** 1Service de Médecine Interne, Hôpital Estaing, CHU de Clermont-Ferrand, 63000 Clermont-Ferrand, France; 2Université Clermont Auvergne, CHU, CNRS, Clermont Auvergne INP, Institut Pascal, UMR 6602, 63000 Clermont-Ferrand, France; 3Service de Médecine Interne et Vasculaire, Hôpital Lyon Sud, Hospices Civils de Lyon, 69310 Pierre-Bénite, France; sabine.mainbourg@chu-lyon.fr (S.M.); jean-christophe.lega@chu-lyon.fr (J.-C.L.); 4CNRS, Laboratoire de Biométrie et Biologie Evolutive, UMR 5558, University of Lyon, Université Lyon 1, 69100 Villeurbanne, France; 5Pulmonary Hypertension Research Group, Institut Universitaire de Cardiologie et de Pneumologie de Québec Research Center, Laval University, Québec, QC G1V 4G5, Canada; vicky.mai.1@ulaval.ca (V.M.); steeve.provencher@criucpq.ulaval.ca (S.P.); 6Service d’Accueil des Urgences, Hospices Civils de Lyon, Hôpital Lyon-Sud, 69310 Pierre-Bénite, France; marion.douplat@chu-lyon.fr; 7Research on Healthcare Performance (RESHAPE), INSERM U1290, Université Lyon 1, 69373 Lyon, France; 8Groupe d’Etude Multidisciplinaire des Maladies Thrombotiques, Hospices Civils de Lyon, 69500 Bron, France

**Keywords:** COVID-19, critical care, meta-analysis, risk factors, venous thromboembolism

## Abstract

Venous thromboembolism (VTE) in patients with COVID-19 in intensive care units (ICU) is frequent, but risk factors (RF) remain unidentified. In this meta-analysis (CRD42020188764) we searched for observational studies from ICUs reporting the association between VTE and RF in Medline/Embase up to 15 April 2021. Reviewers independently extracted data in duplicate and assessed the certainty of the evidence using the GRADE approach. Analyses were conducted using the random-effects model and produced a non-adjusted odds ratio (OR). We analysed 83 RF from 21 studies (5296 patients). We found moderate-certainty evidence for an association between VTE and the D-dimer peak (OR 5.83, 95%CI 3.18–10.70), and length of hospitalization (OR 7.09, 95%CI 3.41–14.73) and intubation (OR 2.61, 95%CI 1.94–3.51). We identified low-certainty evidence for an association between VTE and CRP (OR 1.83, 95% CI 1.32–2.53), D-dimer (OR 4.58, 95% CI 2.52–8.50), troponin T (OR 8.64, 95% CI 3.25–22.97), and the requirement for inotropic drugs (OR 1.67, 95% CI 1.15–2.43). Traditional VTE RF (i.e., history of cancer, previous VTE events, obesity) were not found to be associated to VTE in COVID-19. Anticoagulation was not associated with a decreased VTE risk. VTE RF in severe COVID-19 correspond to individual illness severity, and inflammatory and coagulation parameters.

## 1. Introduction

Severe acute respiratory syndrome coronavirus 2 (SARS-CoV-2) is responsible for an outbreak of respiratory disease called Coronavirus Infectious disease 2019 (COVID-19) [1], which is now a worldwide global burden for public health. Severe cases of COVID-19 are characterized by sepsis-related coagulopathy [2], platelet activation, and endothelial dysfunction [3]. The reported incidence of venous thromboembolic events (VTE) in COVID-19 is highly variable, suggesting that individual risk factors influence the thrombotic risk. We recently demonstrated that patients admitted to the intensive care units (ICU) had higher risk of venous thromboembolism (VTE) than patients admitted in general ward [4] with an estimated prevalence of 23%. A recent systematic review also suggested that VTE were more frequent amongst ICU patients hospitalised for COVID-19 compared to other infections of similar severity [5].

Given the increased risk of VTE, several clinical trials evaluated the role of intermediate dose thromboprophylaxis and high dose anticoagulation [6], showing no difference in survival to hospital discharge or thrombosis prevention [7]. While the American Society of Hematology recommends the use of an early prophylactic-intensity anticoagulation for patients with severe COVID-19 with no confirmed VTE [8], high dose anticoagulation has been associated to an increase of major bleeding [9]. These results might be due to the lack of selection criteria or risk stratification for VTE at inclusion. Thus, there is a true urgency to identify risk factors for VTE in COVID-19 to help physicians with deciding the form of anticoagulant management in their current practice.

## 2. Materials and Methods

We followed the MOOSE (Meta-Analysis of Observational Studies in Epidemiology) guidelines; the Cochrane Prognosis Methods Group guidelines [10]; the Grading of Recommendations, Assessment, Development and Evaluation (GRADE) Working Group handbook [11]; and the PRISMA (Preferred Reporting Items for Systematic reviews and Meta-Analyses) guidelines [12] during all stages of study design, implementation, and reporting of this systematic review and meta-analysis. The protocol was recorded on PROSPERO (CRD42020188764).

### 2.1. Search Strategy

There was a Medline and Embase literature search from 1 January 2020 to 15 April 2021. We aimed to collect all published observational studies reporting VTE in COVID-19. We used the following keywords (including MeSH): coronavirus, severe acute respiratory syndrome coronavirus 2, novel coronavirus, nCoV, 2019-CoV, COVID-19, thrombosis, venous thromboembolism, and pulmonary embolism. The search string is available at https://www.crd.york.ac.uk/PROSPERO/display_record.php?ID=CRD42020188764 (accessed on 6 December 2021).

Additionally, we manually searched relevant articles on the websites of major medical journals (i.e., the New England Journal of Medicine, Journal of the American Medical Association, Lancet, Lancet Haematology, British Medical Journal, Blood, Journal of the American Journal of Cardiology, Circulation, Journal of Thrombosis and Haemostasis, Thrombosis and Haemostasis, and Thrombosis Research) as well as the bibliographies of each included article. We did not retain non-English papers, abstracts, and pre-print papers that were not peer-reviewed.

### 2.2. Study Selection

The inclusion criteria were: (i) cohort studies of at least 10 patients in ICU, (ii) patients with positive reverse transcriptase polymerase chain reaction COVID-19 sample or with suggestive radiologic findings on computed chest tomography, (iii) available rate of documented VTE as defined by the investigators, and (iv) available rate of exposures as tested by investigators, including:-Demographic characteristics: age, male sex;-Patient history factors: body-mass-index (BMI) and obesity (BMI ≥ 30 kg·m^−2^), alcohol and tobacco use, history of cancer, history of previous VTE, several medical conditions (including cardiac disease, cerebrovascular disease, chronic kidney disease, chronic liver disease, chronic lung disease, diabetes mellitus, immunodeficiency, human immunodeficiency virus (HIV) infection, hypertension and use of angiotensin convertor enzyme (ACE) inhibitors);-Therapeutic management and disease severity markers: Acute Physiology and Chronic Health Evaluation II (APACHE II), Simplified Acute Physiology Score II (SAPS-II), Sepsis-related Organ Failure Assessment score (SOFA), Therapeutic Intervention Scoring Systemc-10 (TISS-10), arterial oxygen partial pressure/fractional inspired oxygen (PaO2/FiO2), requirement and duration of use of neuromuscular blockers and inotropic drugs, length of hospital stay and disease duration, acute kidney injury and requirement for renal replacement therapy, myocarditis, requirement for extracorporeal membrane oxygenation (ECMO), use of anticoagulant treatment (including prophylactic, intermediate and therapeutic dosage);-Inflammation biomarkers: C-reactive protein (CRP), ferritin, fibrinogen, interleukin-6 (IL-6) and procalcitonin.-Coagulation tests: activated partial thromboplastin time (APTT), D-dimers, Prothrombin Time (PT), score for disseminated intravascular coagulation (DIC) from the International Society on Thrombosis and Haemostasis (ISTH);-Haematological parameters: complete blood count (including: leukocytes, lymphocytes, neutrophils, haemoglobin, and platelets);-Biological markers of organ dysfunction: alanine aminotransferase (ALT), aspartate aminotransferase (AST), creatine kinase serum level, creatinine level, lactate dehydrogenase (LDH), N-terminal pro-brain natriuretic peptide (NT-proBNP), troponin T, and troponin I level.

Two reviewers (H. L. and J-C. L.) independently applied these criteria to the titles and abstracts of all citations obtained. If pertinent, each reviewer retrieved and explored the complete articles to make a final decision about their inclusion in the meta-analysis. In case of disagreement, a third reviewer (S. M.) was consulted to reach a consensus. Throughout this process, reviewers were blinded to author and journal names. In case of cohorts reported in multiple papers, the analysis was limited to the largest cohort, unless the necessary data had appeared only in another paper. A log of reasons for rejection of citations identified from the searches was kept. In case of doubt or apparent inconsistency, a reviewer (H. L.) would contact the corresponding author to remove any ambiguity.

### 2.3. Quality Assessment

The quality and risk of bias were assessed using the Quality In Prognostic Studies (QUIPS) tool as recommended by the Cochrane Prognosis Methods [10]. Two reviewers (H.L. and J.-C.L.) independently evaluated the risk of bias for each item. If needed, a discussion was planned to reach a consensus. Risk of publication bias was additionally assessed visually using funnel plots and an Egger’s test (using a threshold at *p* < 0.05). We assumed that the effect of publication bias should be minor if the plot of the magnitude of the effect size in each study versus its precision estimate (i.e., standard error) showed a roughly symmetrical funnel shape (if number of studies >10).

### 2.4. Outcomes

The main outcomes were VTE, as defined by the investigators, including deep vein thrombosis (DVT) or pulmonary embolism (PE). We investigated all the prognostic factors tested in individual studies, including the clinical characteristics, biological testing, disease severity markers, and the drug or procedure exposures.

### 2.5. Statistical Analysis

We standardized the units of measurement for each prognostic factor, unifying the direction of the predictors, adjusting the weights of the studies, and calculating crude-effect estimates when not provided. When possible, we performed meta-analysis of all the risk factors to assess their association with VTE occurrence, when reported by more than one study. For a dichotomous outcome, we presented the effect estimate as an OR and the corresponding 95% confidence interval (CI). We conducted a meta-analysis of associations using the generic inverse variance-based method and random-effects model to produce an overall measure of association. For a continuous outcome, we estimated the standardized mean difference. The standardized mean difference was re-expressed as an OR using the method of Chinn et al. [13]. The mean and standard variance were estimated from the medians, and the sample size if needed [14]. A sensitivity analysis was performed using the Bonferroni correction to control the type 1 error (i.e., false positive results related to multiple statistical tests). We explored the consistency of the associations between the results of our meta-analysis. All analyses were performed with R (R Foundation for Statistical Computing, Vienna, Austria).

### 2.6. Role of the Funding Source

No funding was received to perform this meta-analysis. All authors had full access to all the data and the final responsibility to submit for publication. The authors declare no conflict of interest related to this study.

### 2.7. Data Availability

All data relevant to the study are included in the article; raw data will be made available upon reasonable request.

## 3. Results

This section may be divided by subheadings. It should provide a concise and precise description of the experimental results and their interpretation, as well as the experimental conclusions that can be drawn.

### 3.1. Literature Search and Study Characteristics

A total of 5690 articles were retrieved using the search terms. After reviewing titles and abstracts, 75 articles were selected for full text eligibility. Overall, 12 retrospective and 9 prospective studies (5296 patients) [15,16,17,18,19,20,21,22,23,24,25,26,27,28,29,30,31,32,33,34,35] were included in the meta-analysis (Figure 1).

The main characteristics of the included studies are described in Table 1.

Risk factors for PE (*n* = 3), DVT (*n* = 5), or both (*n* = 13) were assessed. In 10 studies, VTE incidence was evaluated as routine care, whereas in 11 studies VTE was evaluated only in case of clinical suspicion. The median number of patients was 81 (range 16–3239). The median follow-up duration was 23 days (range 14–28). The included studies took place in France (*n* = 6), the United States of America (*n* = 5), the United Kingdom (*n* = 3), China (*n* = 2), and other various counties (*n* = 5). Six studies were multicentric.

### 3.2. Risk of Bias

As shown in Table 2, a serious risk of bias was present in all selected studies, with a moderate–high risk of bias in at least one of the six domains of the QUIPS tool. The risk of confounding factors was also considered as moderate–high for all studies.

Conversely, the visual inspection of the funnel plot and the Egger’s regression (*p* > 0.05) test were not in favour of publication bias for the variables reported in >10 studies (Appendix A).

### 3.3. Identification of Risk Factors for VTE

Figure 2 summarizes the main risk factors identified in the present work.

#### 3.3.1. Demographic and Past History Factors

We found no association between the risk of VTE and age [15,17,18,19,20,21,23,25,26,27,28,29,30,31,32,33,35], male sex [15,17,19,20,21,22,23,25,26,27,28,29,30,31,32,35], body mass index (OR 1.04, 95% CI 0.85–1.25) [15,17,19,20,21,23,25,27,28,29,31,32,33] or obesity (OR 1.41, 95% CI 0.64–3.09) [17,27,28,32], history of previous VTE (OR 0.90, 95% CI 0.27–2.98) [17,19,21,23,35], cancer (0.84, 95% CI 0.51–1.38) [15,17,19,23,25,29,31,35], and tobacco consumption [15,25,26,27,28,35].

Furthermore, we found no association between numerous medical conditions and risk of VTE, including cerebrovascular disease [21,35], chronic kidney disease [15,19,21,25,26,28,31,35], chronic respiratory disease—including asthma and COPD [15,19,21,25,26,28,29], diabetes mellitus [15,17,19,21,25,26,27,28,29,31,32,35], hypertension [15,17,19,21,26,27,28,29,32,35], immunodeficiency [15,25,28], and use of ACE inhibitors [19,29].

Conversely, we identified low-certainty evidence for an association between a decreased risk of any VTE and history of cardiovascular disease (OR 0.32, 95% CI 0.10–0.99) [15,17,19,21,25,29,31,32,35], and congestive heart failure (OR 0.57, 95% CI 0.35–0.95) [15,26,28,31].

#### 3.3.2. Therapeutic Management and Disease Severity

We identified an association between VTE risk and requirement for intubation [15,19,20,21,25,29,31] (OR 2.61, 95% CI 1.94–3.51), duration of neuromuscular blockade [31] (OR 2.64, 95% CI 1.57–4.47) with moderate-certainty evidence, and low-certainty evidence for an association between VTE risk and the use of inotropic drugs (OR 1.67, 95% CI 1.15–2.43, low-certainty evidence) [15,17,19,20,21,25,29].

The requirements for ECMO [19,27,28,29,31], the PaO2/FiO2 ratio [17,20,23,31], the duration of mechanical ventilation [23,25], and renal replacement therapy [15,17,19,21,31] were not found to be associated with the risk of VTE with low-certainty evidence. The SOFA score [17,21,23] was associated to VTE risk (OR 3.69, 95% CI 1.22–11.2, low-certainty evidence) when the score was established the day of VTE screening (single study). APACHE-II was associated with the risk of VTE in a single study [30] (OR 1.76, 95% CI 1.04–3.00, very low-certainty evidence).

We found no association between the use of anticoagulation regardless of the dosage used and risk of VTE [16,21,23,24,28,30,32] with low-certainty evidence. We identified moderate-certainty evidence for an association between VTE risk and the length of stay in hospitalization (OR 7.09, 95% CI 3.41–14.73) [19,25], whereas the disease duration before hospitalization [19,26,35], the disease duration before admission to the intensive care units [17,26], the length of stay in the intensive care unit [19,28,30], or the disease duration before VTE screening [17,23] were not found to be associated with the risk for VTE. In contrast, we identified low-certainty evidence for an association between the time from intensive care unit (ICU) admission to VTE screening [17,33] with a decreased risk of VTE (OR 0.29, 95% CI 0.12–0.70).

#### 3.3.3. Inflammation Biomarkers

We identified low-certainty evidence for an association between VTE risk and CRP [15,19,20,22,23,25,26,27,29,30,32,35] (OR 1.83, 95% CI 1.32–2.53) and procalcitonin levels [15,19,26,35] (OR 2.61, 95% CI 1.05–6.36). Furthermore, we found an association between IL-6 levels and the risk of VTE (OR 0.58 95% CI 0.37–0.97) with low-certainty evidence [15,22,26]. Conversely, we found no association between the risk of VTE and the fibrinogen [15,17,19,20,21,22,23,26,27,28,29,30,34] nor ferritin levels [15,19,23,26,27,30,32].

#### 3.3.4. Coagulation Tests

We found moderate to low-certainty evidence for an association between VTE and the D-dimer levels (OR 4.58, 95% CI 2.52–8.50, low-certainty evidence) [15,17,18,19,20,21,22,23,25,26,27,28,29,30,32,34,35], D-dimer peak (OR 5.83, 95% CI 3.18–10.7, moderate-certainty evidence) [26,32,33], and D-dimer at VTE screening (OR 5.14, 95% CI 1.95–13.6, low-certainty evidence) [28,33]. We found no evidence for the PT value [18,20,27,30,32,35], PT percentage [21,29], APTT value [18,20,27,32,35], and APTT ratio [19,29].

#### 3.3.5. Haematological Parameters

In a single study, an association was found between the platelets peak count and a decreased risk of VTE with low certainty-evidence (OR 0.25, 95% CI 0.12–0.55). We found no association between the risk of VTE and the haemoglobin levels [15,18,19,25,26,27,30,35], leukocytes count [15,18,19,25,26,27,29,30,35], lymphocytes count [18,25,27,29,30,35], and platelets count [15,17,18,19,20,21,23,25,26,27,29,30,35].

#### 3.3.6. Biological Markers of Organ Dysfunction

We found no association between VTE and the creatinine levels [22,25,26,27,29,35], hepatic enzymes levels [25,26,27,35], creatine kinase levels [27], lactate dehydrogenase levels [23,25,26,27,30,32,35], and N-terminal pro-brain natriuretic peptide levels [27,29]. Low-certainty evidence was found with troponin T [26] (OR 8.64, 95% CI 3.25–22.97) in a single study, whereas no evidence was found for troponin I [27,30,35].

Levels of evidence and the estimated OR are summarized in Table 3 and Table 4.

### 3.4. Sensitivity Analysis

We performed 92 statistical tests. Using the Bonferonni correction, only the hospital stay duration (*p* < 0.0001), peak of D-dimer (*p* < 0.0001), D-dimer (*p* < 0.0001), troponin T (*p* < 0.0001), and mechanical ventilation (*p* < 0.0001) remained statistically significant.

## 4. Discussion

In the present meta-analysis that encompasses 5296 ICU patients from 21 observational studies, we identified low –to moderate-certainty evidence for the risk factors for VTE, including the COVID-19 severity (i.e., requirement for mechanical ventilation, use of inotropic drugs, length of hospital stay, and SOFA score at VTE screening) and biological parameters (i.e., D-dimer levels, DIC ISTH criteria, CRP level, troponin T, procalcitonin, thrombocytopenia). Some risk factors previously reported were not confirmed by our analysis, including an older age, female sex, obesity [36,37], and a medical history of VTE or cancer [38,39]. Interestingly, the use of anticoagulants at any dose was not associated with a different risk of thrombosis in the univariate analysis. The level of evidence for VTE risk factors was most prominent for the D-dimer and CRP levels, followed by procalcitonin, IL-6, and markers of severity (i.e., mechanical ventilation, inotrope, or neuromuscular blockade duration). The negative correlation between VTE, congestive heart failure, and the history of cardiovascular disease might be explained by the anticoagulant use at admission for these patients, which would hamper interpretation of these results.

Taken together, the present meta-analysis provides more precise estimates of the association between individual risk factors and the risk of VTE, which may help identifying ICU patients at lower and higher risk of VTE. Three randomized trials in ICU patients reported the absence of efficacy of therapeutic anticoagulation or intermediate-dose thromboprophylaxis compared to the standard dose thromboprophylaxis [6,7,40], whereas therapeutic anticoagulation was associated with increased survival in non-critically ill patients hospitalized for COVID-19 [41]. In patients with mild COVID-19 that are not mechanically ventilated, therapeutic heparin use was not associated with a decreased risk of death, intubation, ICU admission, or thromboembolism when compared to prophylactic anticoagulation [42].

These trials included patients irrespective of their D-dimer levels or inflammatory parameters. Our study identified the candidate risk factors for VTE in ICU patients with COVID-19, which is crucial for VTE screening and prevention. Indeed, the stratifications of VTE risk may increase the benefit–risk ratio of primary prevention of thrombosis using intermediate or therapeutic anticoagulant dose. This concept is supported by a retrospective monocentric study including critically ill COVID-19 patients, suggesting that this subgroup of patients with elevated D-dimer levels requiring mechanical ventilation—two risk factors identified in the present meta-analysis—may benefit from therapeutic anticoagulation [43]. Although subgroup analysis from recent trials did not show any interaction between D-dimer levels and the efficacy of full-dose anticoagulation, critically ill COVID-19 patients are likely to represent a highly heterogeneous group of patients with a diverse risk of VTE and bleeding. We thus speculate that the individual risk stratification could be useful to identify patients susceptible to benefit from systematic screening or anticoagulation intensification.

Interestingly, we confirmed the increased risk of VTE amongst patients with high inflammation biomarkers, whereas “traditional risk factors” for thrombosis (i.e., cancer, history of VTE, obesity [44]) were not associated with VTE. These observations support the concept that thrombosis in COVID-19 may result from mechanisms implicating pulmonary inflammation, intravascular coagulopathy associated with D-dimer elevation [45], endothelial dysfunction, and immunothrombosis [46] (Figure 2). In our meta-analysis, the best tool to assess sepsis-related coagulopathy seemed to be CRP elevation and, to a lesser extent, procalcitonin—with regards to the risk of a false positive result for the latter. The clinical disease-related severity markers were the consequence of both sepsis-related coagulopathy and systemic inflammation. Accordingly, targeting inflammation may be considered as a potential additional therapeutic target to prevent COVID-19-related thrombotic complications [47].

A potential limitation of the present meta-analysis was the variability in the study sample, study design, patients’ characteristics and management, and the definition of VTE events. In addition, many estimates were associated with large confidence intervals despite the increased statistical power provided by the meta-analysis. Conversely, because of the multiple statistical tests, we could not exclude fortuitous association (i.e., false positive results related to type 1 error). This may explain the protective effect of the presence of cardiovascular disease, or the effect of IL-6 levels and cardiovascular disease on VTE risk. These factors were non-significant after the Bonferroni correction. Nonetheless, this study is unique in many aspects, including its comprehensiveness, the novelty of its findings, and its rigorous methods. Finally, we produced a crude estimate of the risk factors, with a high probability of collinearity between variables between the inflammatory and coagulation parameters [47]. Multicentric studies using a multivariable model are thus warranted to validate the independent variables predicting VTE occurrence.

## 5. Conclusions

The present meta-analysis identified the relevant risk factors of VTE in patients admitted to the ICU for severe COVID-19, mainly inflammation profile and illness severity. These findings may help identify individual patients at higher risk of VTE, inform experts developing risk assessment models to risk-stratify patients, and ultimately, improve patients’ outcomes through optimal screening and management strategies.

## Figures and Tables

**Figure 1 ijerph-18-12944-f001:**
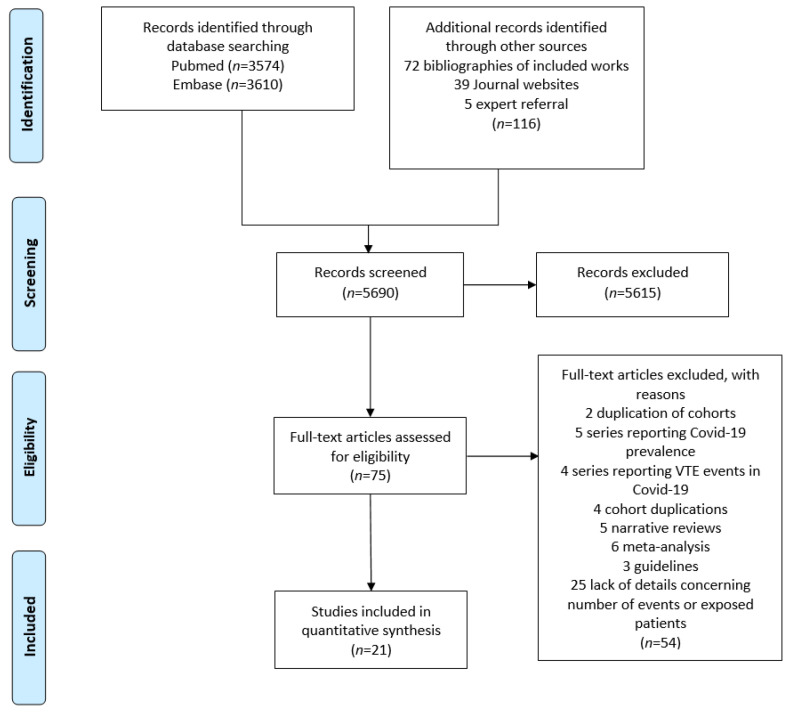
Flow diagram of the literature search and study inclusion.

**Figure 2 ijerph-18-12944-f002:**
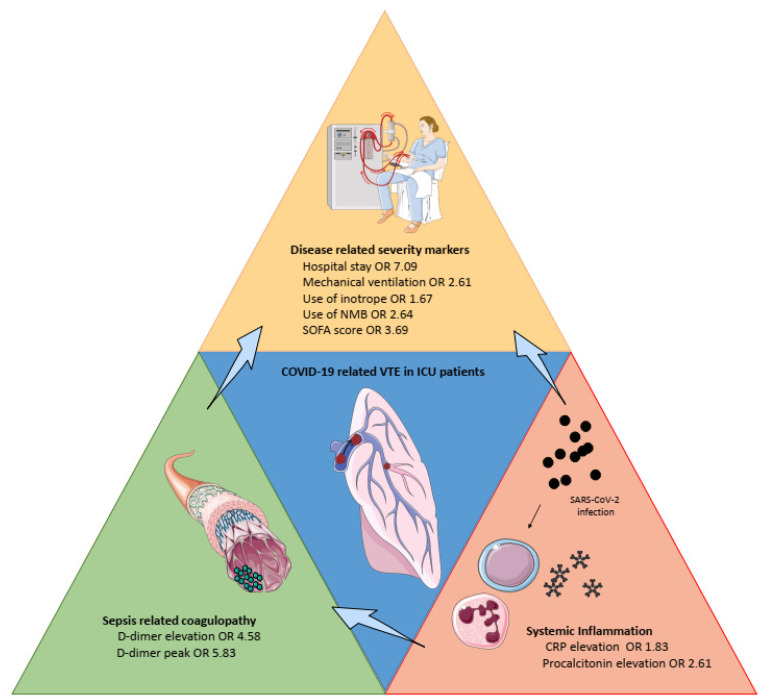
Overview of the main risk factors for venous thromboembolism in severe COVID-19. CRP: C-reactive protein, ICU: intensive care unit, NMB: neuromuscular blockade, SOFA: Sepsis-related Organ Failure Assessment.

**Table 1 ijerph-18-12944-t001:** Main characteristics of included studies.

Study	Country	Nb	FU	Design	Diagnostic of COVID-19	Male%	Mean Age (y)	Diagnosis of VTE	Type of VTE Event	VTE Rate	Data Collection
Al-Samkari [15]	USA	3239	27	P, C, M	RT-PCR	64.5%	61	NA	DVT, PE	6%	ICU admission
Bellmunt-Montoya [16]	Spain	227	7	P, NC, M	RT-PCR	77%	62	AS	DVT, PE	26.5%	ICU admission
Contou [17]	France	26	NA	R, NC, m	RT-PCR	79%	61	ST	PE	62%	Day of CTPA
Cui [18]	China	81	14	R, NA, m	RT-PCR and CT-scan	46%	60	AS	DVT, PE	25%	NA
Desborough [19]	UK	66	28	R, C, m	Antigen test	73%	59	ST	DVT, PE	15%	ICU admission
Dujardin [20]	Netherlands	127	NA	R, C, m	RT-PCR	77%	62	AS	DVT, PE	41%	ICU admission
Fraissé [21]	France	92	NA	R, C, m	NA	79%	61	ST	DVT, PE	40%	ICU admission
Gibson [22]	USA	72	NA	P, NA, m	RT-PCR	79%	64	AS	DVT	17%	Hospital admission
Grandmaison [23]	Switzerland	29	NA	P, NC, m	RT-PCR	62%	66	AS (DVT) and ST (PE)	DVT, PE	59%	NA
Helms [24]	France	179	80	P, C, M	RT-PCR	73%	62	ST	DVT, PE	32%	NA
Hippensteel [25]	USA	91	26	R, NC, m	NA	58%	56	ST	DVT, PE	26%	Hospital admission
Maatman [26]	USA	109	NA	R, C, M	RT-PCR	62%	61	ST	DVT, PE	16%	NA
Mirsadraee [27]	UK	72	32	R, C, m	RT-PCR	74%	52	ST	DVT, PE	58%	NA
Mueller-Peltzer [28]	Germany	16	23	R, NC, m	RT-PCR	73%	60	AS	PE	45%	Mixed
Nahum [29]	France	34	NA	P, C, m	RT-PCR and CT-scan	78%	62	AS	DVT	79%	NA
Shah [30]	UK	187	20	R, C, M	RT-PCR and CT-scan	68%	57	ST	DVT, PE	35%	< 48 h ICU admission
Soumagne [31]	France, Belgium	375	28	NA, C, M	RT-PCR	77%	62	ST	PE	36%	NA
Torres-Machorro [32]	Mexico	30	<9	P, C, m	NA	77%	62	AS	DVT	30%	NA
Trigonis [33]	USA	45	NA	R, NC, m	NA	NA	61	ST	DVT	42%	VTE screening
Voicu [34]	France	56	10	P, C, m	NA	75%	NA	AS	DVT	46%	NA
Zhang [35]	China	143	19	R, C, m	RT-PCR	51%	63	AS	DVT, PE	46%%	NA

AS: asymptomatic screening; C: consecutive, CTPA: computed tomography pulmonary angiogram; CT-scan: computed tomography scanner; DVT: deep vein thrombosis; FU: mean follow-up (days); ICU: intensive care unit; M: multicentric; m: monocentric; NA: Not assessable; Nb: number of patients; NC: non-consecutive; P: prospective; PE: pulmonary embolism; R: retrospective; RT-PCR: reverse transcriptase-polymerase chain reaction; ST: symptomatic testing; UK: United Kingdom; USA: United States of America; VTE: venous thromboembolism; y: year.

**Table 2 ijerph-18-12944-t002:** Risk of bias assessment using QUIPS tool.

Study	Study Participation	Study Attrition	Prognostic Factor Measurement	Outcome Measurement	Study Confounding	Statistical Analysis and Reporting
Al-Samkari [15]						
Bellmunt-Montoya [16]						
Contou [17]						
Cui [18]						
Desborough [19]						
Dujardin [20]						
Fraissé [21]						
Gibson [22]						
Grandmaison [23]						
Helms [24]						
Hippensteel [25]						
Maatman [26]						
Mirsadraee [27]						
Mueller-Peltzer [28]						
Nahum [29]						
Shah [30]						
Soumagne [31]						
Torres-Machorro [32]						
Trigonis [33]						
Voicu [34]						
Zhang [35]						

Green indicates low risk of bias; red indicates high risk of bias; yellow indicates moderate risk of bias.

**Table 3 ijerph-18-12944-t003:** Evidence summary for risk factors related to venous thromboembolism (continuous variables).

Variable		Certainty Assessment Domains						Overall Certainty	SMD 95%CI	OR 95%CI
	N	Study Design	Risk of Bias	Indirectness	Inconsistency	Imprecision	Publication Bias			
Demographic factors										
Age	17	Obs	S *	NS	S	S	Undetect	ꚛ○○○	0.11 (−0.09; 0.30)	1.22 (0.85; 1.72)
BMI	13	Obs	S *	NS	NS	NS	Undetect	ꚛꚛꚛ○	0.02 (−0.08–0.12)	1.04 (0.86–1.25)
Biological markers of organ dysfunction										
ALAT	4	Obs	S *	NS	S	S	Undetect	ꚛ○○○	0.24 (−0.07; 0.56)	1.55 (0.87; 2.75)
ASAT	3	Obs	S *	NS	S	S	Undetect	ꚛꚛ○○	0.25 (−0.09; 0.60)	1.59 (0.85; 2.97)
Creatinine	6	Obs	S *	NS	S	NS	Undetect	ꚛꚛ○○	−0.08 (−0.39; 0.24)	0.87 (0.49; 1.54)
Creatine kinase	1	Obs	S *	NS	NS	S	Undetect	ꚛ○○○	−0.25 (−0.73–0.23)	0.64 (0.27–1.53)
LDH	6	Obs	S *	NS	S	S	Undetect	ꚛꚛ○○	0.44 (−0.03; 0.91)	2.23 (0.95; 5.19)
LDH (peak)	1	Obs	S *	NS	NS	S	Undetect	ꚛꚛ○○	0.15 (−0.3–0.6)	1.32 (0.58–2.95)
NT-proBNP	2	Obs	S *	NS	NS	S	Undetect	ꚛꚛ○○	−0.06 (−0.66–0.54)	0.90 (0.30–2.64)
Troponin I	3	Obs	S *	NS	S	S	Undetected	ꚛ○○○	0.94 (−0.10; 1.98)	5.49 (0.83; 36.28)
Troponin T	1	Obs	S *	NS	NS	S	Detected	ꚛꚛ○○	1.19 (0.65; 1.73)	8.64 (3.25; 22.97)
Scores										
APACHEII	1	Obs	S *	NS	NS	S	Detected	ꚛ○○○	0.31 (0.02; 0.61)	1.76 (1.04; 3.00)
DIC ISTH	1	Obs	S *	NS	NS	S	Detected	ꚛ○○○	0.44 (0.15; 0.73)	2.23 (1.31; 3.79)
SAPSII	3	Obs	S *	NS	NS	S	Undetect	ꚛꚛ○○	0.13 (−0.22; 0.48)	1.27 (0.68; 2.37)
SOFA	3	Obs	S *	NS	S	S	Undetect	ꚛ○○○	0.24 (−0.3–0.79)	1.55 (0.58–4.17)
SOFA at VTE screening	1	Obs	S *	NS	NS	S	Detected	ꚛꚛ○○	0.72 (0.11–1.33)	3.69 (1.22–11.2)
SOFA at intubation	1	Obs	S *	NS	NS	S	Undetect	ꚛꚛ○○	0.33 (−0.26–0.93)	1.83 (0.62–5.4)
TISS 10	1	Obs	S *	NS	NS	S	Undetect	ꚛꚛ○○	−0.66 (−1.68; 0.36)	0.30 (0.05–1.93)
Coagulation markers										
aPTT ratio	2	Obs	S *	NS	S	S	Detected	ꚛ○○○	0.27 (−4.07; 4.60)	1.63 (0.00; 2.76)
aPTT	5	Obs	S *	NS	NS	S	Undetect	ꚛꚛ○○	0.23 (−0.15–0.60)	1.5 (0.76–2.99)
PT (absolute)	6	Obs	S *	NS	S	S	Undetect	ꚛ○○○	0.06 (−0.34; 0.45)	1.11 (0.54; 2.27)
PT (%)	2	Obs	S *	NS	S	S	Undetect	ꚛ○○○	−0.11 (−1.35; 1.13)	0.82 (0.09; 7.69)
D-Dimer	17	Obs	S *	NS	S	NS	Undetect	ꚛꚛ○○	0.84 (0.51; 1.18)	4.58 (2.52; 8.50)
D-Dimer (peak)	3	Obs	S *	NS	NS	NS	Undetec	ꚛꚛꚛ○	0.97 (0.64; 1.31)	5.83 (3.18; 10.7)
D-Dimer at VTE screening	2	Obs	S *	NS	NS	S	Undetec	ꚛꚛ○○	0.90 (0.37; 1.44)	5.14 (1.95; 13.6)
Inflammation makers										
CRP	12	Obs	S *	NS	S	NS	Undetect	ꚛꚛ○○	0.32 (0.14–0.51)	1.83 (1.32–2.53)
CRP (peak)	1	Obs	S *	NS	NS	S	Undetect	ꚛꚛ○○	−0.01 (−0.46; 0.43)	0.98 (0.44; 2.17)
Ferritin	7	Obs	S *	NS	S	S	Detected	ꚛ○○○	0.35 (−0.30; 1.00)	1.89 (0.58; 6.13)
Ferritin (peak)	1	Obs	S *	NS	NS	S	Undetect	ꚛꚛ○○	0.00 (−0.45–0.45)	0.99 (0.44–2.25)
Fg	13	Obs	S *	NS	NS	S	Undetected	ꚛꚛ○○	−0.02 (−0.16; 0.19)	0.96 (0.75–1.41)
Fg (peak)	1	Obs	S *	NS	NS	S	Undetected	ꚛꚛ○○	−0.23 (−0.73–0.27)	0.66 (0.26–1.63)
IL−6	3	Obs	S *	NS	NS	S	Undetect	ꚛꚛ○○	−0.28 (−0.55; −0.02)	0.58 (0.37; 0.97)
PCT	4	Obs	S *	NS	S	S	Undetect	ꚛ○○○	0.53 (0.03; 1.02)	2.61 (1.05; 6.36)
PCT (peak)	1	Obs	S *	NS	NS	S	Undetect	ꚛꚛ○○	0.41 (−0.0; 0.87)	2.08 (0.89; 4.87)
Haematological parameters										
Haemoglobin	8	Obs	S *	NS	NS	S	Undetect	ꚛꚛ○○	0.02 (−0.10; 0.13)	1.03 (0.83–1.27)
Leukocytes	9	Obs	S *	NS	S	S	Undetect	ꚛꚛ○○	0.34 (−0.03; 0.7)	1.84 (0.94; 3.59)
Lymphocytes	6	Obs	S *	NS	S	S	Detected	ꚛ○○○	−0.23 (−0.52; 0.06)	0.66 (0.39; 1.11)
Neutro/lympho ratio	3	Obs	S *	NS	S	NS	Detected	ꚛ○○○	−0.31 (−1.64; 1.02)	0.57 (0.05; 6.38)
Platelets	13	Obs	S *	NS	NS	NS	Undetect	ꚛꚛꚛ○	0.04 (−0.07; 0.15)	1.08 (0.88; 1.32)
Platelets (peak)	1	Obs	S *	NS	NS	S	Detected	ꚛꚛ○○	−0.76 (−1.19; −0.33)	0.25 (0.12; 0.55)
Disease severity markers										
NMB duration	1	Obs	S *	NS	NS	NS	Detected	ꚛꚛ○○	0.54 (0.25–0.83)	2.64 (1.57–4.47)
PaO2/FiO2	4	Obs	S *	NS	NS	S	Undetect	ꚛꚛ○○	−0.13 (−0.41; 0.15)	0.79 (0.48; 1.31)
Time factors										
Disease onset to hospitalization	2	Obs	S *	NS	NS	S	Undetect	ꚛꚛ○○	0.09 (−0.27–0.44)	1.18 (0.62–2.23)
Disease onset to ICU	2	Obs	S *	NS	NS	S	Undetect	ꚛꚛ○○	−0.20 (−0.57; 0.17)	0.70 (0.36; 1.36)
Time to VTE screening	2	Obs	S *	NS	S	S	Undetect	ꚛ○○○	−0.05 (−0.83; 0.72)	0.91 (0.22; 3.71)
VTE screening in ICU	2	Obs	S *	NS	NS	S	Detected	ꚛꚛ○○	−0.68 (−1.17; −0.19)	0.29 (0.12; 0.70)
Hospital stay duration	2	Obs	S *	NS	NS	NS	Undetect	ꚛꚛꚛ○	1.08 (0.68–1.48)	7.09 (3.41–14.73)
MV duration	2	Obs	S *	NS	NS	S	Undetect	ꚛꚛ○○	0.39 (−0.06; 0.84)	2.02 (0.90; 4.55)
ICU stay duration	3	Obs	S *	NS	S	S	Detected	ꚛ○○○	0.21 (−0.78; 1.20)	1.46 (0.24; 8.81)

ALAT: alanine aminotransferase; APACHE: Acute Physiology And Chronic Health Evaluation; APTT: activated partial thromboplastin time; ASAT: aspartate aminotransferase; BMI: body mass index; CRP: C reactive protein; DIC-ISTH: Disseminated Intravascular Coagulation from the International Society on Thrombosis and Haemostasis; ICU: intensive care unit; IL-6: interleukin 6; LDH: lactate dehydrogenase; MV: mechanical ventilation; N: number of studies; NS: not serious; NT-proBNP: N-terminal pro-brain natriuretic peptide; PaO2/FiO2: arterial oxygen partial pressure/fractional inspired oxygen; PT: prothrombin time; S: serious; SAPSII: Simplified Acute Physiology Score II; SOFA: Sepsis-related Organ Failure Assessment; TISS10: Therapeutic Intervention Scoring System; Undetec: undetected; VTE: venous thromboembolism; 95% CI: confidence interval 95%; OR: odds ratio; SMD: standardized mean difference; * Certainty in evidence was downgraded for risk of bias, given that confounders were not excluded. ꚛ○○○ indicates very low certainty-evidence, ꚛꚛ○○ indicates low-certainty evidence, ꚛꚛꚛ○ indicates moderate-certainty evidence.

**Table 4 ijerph-18-12944-t004:** Evidence summary for prognostic factors related to venous thromboembolism (discrete variables).

Variable		Certainty Assessment Domains						Overall Certainty	OR 95%CI
	N	Study Design	Risk of Bias	Indirectness	Inconsistency	Imprecision	Publication Bias		
Medical history									
Asthma	2	Obs	S *	NS	NS	S	Undetected	ꚛꚛ○○	0.95 (0.62–1.47)
ACE inhibitor use	2	Obs	S *	NS	NS	S	Undetected	ꚛꚛ○○	0.67 (0.19–2.33)
Alcohol use	1	Obs	S *	NS	NS	S	Undetected	ꚛꚛ○○	0.47 (0.14–1.55)
Cancer	8	Obs	S *	NS	NS	S	Undetected	ꚛꚛ○○	0.84 (0.51–1.38)
Cardiovascular disease	2	Obs	S *	NS	NS	S	Undetected	ꚛꚛ○○	0.32 (0.10–0.99)
Cerebrovascular disease	2	Obs	S *	NS	NS	S	Undetected	ꚛꚛ○○	0.84 (0.26–2.65)
Chronic kidney disease	8	Obs	S *	NS	NS	S	Undetected	ꚛꚛ○○	0.73 (0.50–1.06)
Chronic respiratory disease	7	Obs	S *	NS	NS	S	Undetected	ꚛꚛ○○	0.76 (0.47–1.23)
COPD	3	Obs	S *	NS	NS	S	Undetected	ꚛ○○○	0.42 (0.11–1.61)
Congestive heart failure	4	Obs	S *	NS	NS	S	Undetected	ꚛꚛ○○	0.57 (0.35–0.95)
Coronary artery disease	5	Obs	S *	NS	NS	S	Undetected	ꚛꚛ○○	1.00 (0.56–1.77)
Diabetes mellitus	8	Obs	S *	NS	NS	S	Undetected	ꚛꚛ○○	0.76 (0.52–1.11)
HIV	1	Obs	S *	NS	NS	S	Undetected	ꚛꚛ○○	1.75 (0.68–4.46)
Hypertension	10	Obs	S *	NS	NS	S	Undetected	ꚛꚛ○○	0.85 (0.64–1.11)
Immunodeficiency	3	Obs	S *	NS	NS	S	Undetected	ꚛꚛ○○	1.24 (0.56–2.73)
Myocarditis	1	Obs	S *	NS	NS	S	Undetected	ꚛꚛ○○	1.95 (0.50–7.57)
Previous VTE	5	Obs	S *	NS	NS	S	Undetected	ꚛꚛ○○	0.90 (0.27–2.98)
Smoker (past or current)	3	Obs	S *	NS	NS	S	Undetected	ꚛꚛ○○	0.96 (0.60–1.52)
Smoker (current)	2	Obs	S *	NS	NS	S	Undetected	ꚛꚛ○○	0.67 (0.16–2.83)
Smoker (past)	1	Obs	S *	NS	NS	S	Undetected	ꚛꚛ○○	0.92 (0.26–3.17)
Disease severity markers and therapeutic management									
Acute kidney injury	1	Obs	S *	NS	NS	S	Undetected	ꚛꚛ○○	1.73 (0.44–6.81)
ECMO	5	Obs	S *	NS	NS	S	Undetected	ꚛꚛ○○	0.95 (0.46–1.97)
Inotrope use	7	Obs	S *	NS	NS	S	Undetected	ꚛꚛ○○	1.67 (1.15–2.43)
Mechanical ventilation	7	Obs	S *	NS	NS	NS	Undetected	ꚛꚛꚛ○	2.61 (1.94–3.51)
Neuromuscular blockers	2	Obs	S *	NS	NS	S	Undetected	ꚛꚛ○○	1.41 (0.64–3.09)
Renal replacement	5	Obs	S *	NS	NS	S	Undetected	ꚛꚛ○○	1.39 (0.93–2.10)
Anticoagulation									
No anticoagulation	2	Obs	S *	NS	NS	S	Undetected	ꚛꚛ○○	6.32 (0.73–54.58)
Prophylactic	5	Obs	S *	NS	NS	S	Undetected	ꚛꚛ○○	0.73 (0.46–1.17)
Intermediate	1	Obs	S *	NS	NS	S	Undetected	ꚛꚛ○○	0.70 (0.29–1.70)
Therapeutic	7	Obs	S *	NS	S	S	Undetected	ꚛꚛ○○	1.21 (0.61–2.38)
Demographic factors									
Male sex	16	Obs	S *	NS	NS	S	Undetected	ꚛꚛ○○	1.32 (0.84–2.06)
Obesity	4	Obs	S *	NS	NS	S	Undetected	ꚛꚛ○○	1.41 (0.64–3.09)

ACE: angiotensin converting enzyme; COPD: chronic obstructive pulmonary disease; ECMO: extracorporeal membrane oxygenation; HIV: human immunodeficiency virus; NS: not serious; Obs: observational; S: serious; VTE: venous thromboembolism; * Certainty in evidence was downgraded for risk of bias, given that confounders were not excluded. ꚛ○○○ indicates very low certainty-evidence, ꚛꚛ○○ indicates low-certainty evidence, ꚛꚛꚛ○ indicates moderate-certainty evidence.

## Data Availability

The data presented in this study are available on request from the corresponding author.

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
