# Peer review of "Risk Factors for Venous Thromboembolism in Severe COVID-19: A Study-Level Meta-Analysis of 21 Studies"

_ijerph, 2021, doi:10.3390/ijerph182412944_

Round 1

Reviewer 1 Report

This is a very nice work which illustrates new findings regarding the association of Venous thromboembolism risk factor in patients with COVID-19 in intensive care units.

This study concludes that venous thromboembolism risk factor in severe COVID-19 correspond to individual illness severity, inflammatory and coagulation parameters.

Author Response

We thanks the reviewer for this comment

Reviewer 2 Report

This is a well written paper with appropriate statistical analysis. However, given that there are about 30 different variables compared, spurious correlations can happen just by chance. Although the authors have mentioned it in the limitation of the study, I would recommend that they make corrections to the analysis for multiple comparisons. This would not necessarily rid of type I errors but will attempt to control for it.

Author Response

We thanks the reviewer for this comment. We added in the manuscript a sensitivity analysis using Bonferroni correction. This point was extended in the limitation section.

Reviewer 3 Report

The Authors provide an interesting metaanalysis regarding the association of several risk factor for VTE and VTE incidence in COVID-19 patients admitted to ICU.

The analysis is sound, the results are interesting and the referebce section is updated. However, I feel that the manuscript might be significantly enriched and improved.

In particular, it would be worth commenting upon the risk factors analysed in the 3.3 section that only contains a reference to Fig.2.

Both Fig.2 and all Tables must be better presented in order to improve manuscript quality and easiness in reading it.

Few minor typos run through the manuscript.

Author Response

Thank you for these comments.

The value of biomarkers and their relations was added in the discussion section in regards of the Figure 2.

We modify the Tables to increase the space between SMD and OR. Notably, the Tables were built in accordance of the GRADE guidelines for the variables and the IJERPH guidelines for the form. Please, could you provide further details concerning potential modifications if needed?

For the Figure 2, we changed the color of the police to improve the readability and added the meaning of abbreviations.
